# Role of Lipids in the Pathogenesis of Parkinson’s Disease

**DOI:** 10.3390/ijms25168935

**Published:** 2024-08-16

**Authors:** Shumpei Kamano, Daisaku Ozawa, Kensuke Ikenaka, Yoshitaka Nagai

**Affiliations:** 1Department of Neurology, Kindai University Faculty of Medicine, Osaka-Sayama 589-8511, Osaka, Japan; kamano.shumpei@gmail.com (S.K.); dai18cst@gmail.com (D.O.); 2Department of Neurology, Osaka University Graduate School of Medicine, Suita 565-0871, Osaka, Japan; ikenaka@neurol.med.osaka-u.ac.jp; 3Life Science Research Institute, Kindai University, Osaka-Sayama 589-8511, Osaka, Japan

**Keywords:** Parkinson’s disease, α-synuclein, lipids, glucocerebrosidase, synaptojanin

## Abstract

Aggregation of α-synuclein (αSyn) and its accumulation as Lewy bodies play a central role in the pathogenesis of Parkinson’s disease (PD). However, the mechanism by which αSyn aggregates in the brain remains unclear. Biochemical studies have demonstrated that αSyn interacts with lipids, and these interactions affect the aggregation process of αSyn. Furthermore, genetic studies have identified mutations in lipid metabolism-associated genes such as glucocerebrosidase 1 (GBA1) and synaptojanin 1 (SYNJ1) in sporadic and familial forms of PD, respectively. In this review, we focus on the role of lipids in triggering αSyn aggregation in the pathogenesis of PD and propose the possibility of modulating the interaction of lipids with αSyn as a potential therapy for PD.

## 1. Introduction

Parkinson’s disease (PD) is a neurodegenerative disease in which patients show motor symptoms, such as tremor, bradykinesia, and rigidity, as well as nonmotor symptoms, such as dementia, autonomic dysfunction, and sleep disorders. Dopamine replacement therapy is used as a symptomatic treatment, but there is no treatment available at present that can attenuate the progression of PD [1]. Pathologically, PD is characterized by Lewy bodies (LBs) composed mainly of α-synuclein (αSyn), which are also characteristic features of dementia with Lewy bodies (DLB) [2,3,4]. Genetic studies have shown that missense mutations (A30P, E46K, H50Q, G51D, A53T, and A53E), as well as duplications and triplication mutations in the SNCA gene encoding αSyn, are responsible for familial forms of PD (fPD) [5,6,7,8,9,10,11,12,13]. Genome-wide association studies have also reported that single nucleotide polymorphisms (SNPs) in the SNCA gene are a major risk factor for sporadic PD (sPD) [14,15,16]. Moreover, animal studies have demonstrated that the expression of either wild-type (wt) or mutant αSyn induces neurodegeneration accompanied by the accumulation of αSyn aggregates, as well as motor function deficits [17,18,19]. These lines of evidence strongly suggest that αSyn plays a central role in the pathogenesis of both familial and sporadic forms of PD. Biochemical studies have shown that αSyn aggregates into oligomers and amyloid-like fibrils in vitro under various conditions [20,21]. Moreover, these αSyn oligomers and amyloid-like fibrils have been shown to exert toxicity. However, it is still unknown as to how wt αSyn aggregates in vivo despite its important roles in the pathogenesis of PD.

αSyn is expressed in the brain, which is a lipid-rich tissue [22,23]. Lipids in the brain include fatty acids, triacylglycerols, phospholipids, sterols, and glycolipids. Physiologically, αSyn is suggested to be involved in fatty acid metabolism in the brain [24,25]. Furthermore, αSyn is reported to interact with the lipid membrane. Interestingly, its interaction with lipids is shown to induce the aggregation of αSyn, implying the possible roles of lipids in the pathogenesis of PD [26]. Supporting this hypothesis, mutations in the glucocerebrosidase 1 (GBA1) and synaptojanin 1 (SYNJ1) genes, which are involved in lipid metabolism, have been identified as risk factors for sPD and fPD, respectively. We and others have demonstrated that dysregulation of the substrate lipids by GBA1 and SYNJ1 mutations promotes the aggregation of wt αSyn [27,28,29]. Furthermore, additional genetic risk factors, including phospholipase A2G6 (*PLA2G6*), vacuolar protein sorting 13C (*VPS13C*), chromosome 19 open reading frame 12 (*C19orf12*), and galactosylceramidase (*GALC*), have also been reported [30,31,32,33]. Moreover, using Fourier-transform infrared spectroscopy imaging, we demonstrated the accumulation of lipids in the central region of LBs in the brains of PD patients [34]. In this review, we focus on the association between lipids and αSyn aggregation. Furthermore, we propose that lipids play an important role in the pathogenesis of PD via the aggregation of αSyn.

## 2. Association between Lipids and αSyn Aggregation

Synucleins are a family that comprises the α, β, and γSyns. While αSyn is associated with diseases such as PD and DLB, no disease-associated mutations have been reported for βSyn and γSyn [35]. αSyn is expressed in several tissues, including the brain, erythrocytes, lymphocytes, muscle, kidney, heart, and lung. In the brain, αSyn has been reported to be localized in the presynaptic terminals of neurons [36]. αSyn binds to lipid membranes, particularly to synaptic vesicles in presynaptic terminals [36,37,38,39,40,41,42], and in the rat brain, about 15% of αSyn is in the membrane-bound form [43]. It was also reported that αSyn and βSyn colocalize in the mouse brain and human brain [41,42,44].

αSyn consists of 140 amino acids and is divided into the following three regions: the N-terminal region (amino acids 1–60), the non-amyloid-β component (NAC) region (amino acids 61–95), and the C-terminal region (amino acids 96–140) (Figure 1). αSyn is thought to exist in an unfolded state in aqueous solution [38,45]. On the other hand, αSyn is known to adopt an α-helix structure when bound to lipid membranes in the cellular milieu. Moreover, αSyn is converted to β-sheet-rich aggregates in LBs [34,46,47,48,49,50]. The NAC region is essential for the aggregation of αSyn, which is ultimately converted to a β-sheet-rich structure during the aggregation process [50,51,52]. Deletion of the highly hydrophobic amino acids 71 to 82 in the NAC domain of αSyn abolishes its aggregation [50], and peptides derived from the NAC region alone can form β-sheet-rich fibrils [53], indicating that the NAC domain of αSyn is important for aggregation. The N-terminal region of αSyn, unlike the C-terminal region, is highly conserved among the synuclein species [38,39,40]. The C-terminal region is known to interact with the NAC region, resulting in the suppression of αSyn aggregation [54]. Moreover, the C-terminal region of αSyn also transiently interacts with its N-terminal region to form a compact monomeric state, leading to the suppression of aggregation [55].

The physiological function of αSyn remains poorly understood. αSyn is abundant in presynaptic terminals and is known to colocalize with presynaptic proteins [56]. In addition, the overexpression of αSyn in mice impairs synaptic function [57,58,59,60]. All the mutations in αSyn that cause familial PD are located in the N-terminal lipid membrane-binding domain. Among them, overexpression of the A53T and E46K mutants of αSyn has been reported to inhibit neurotransmitter release [58,61,62]. The A53T mutant has preserved membrane-binding affinity, whereas the E46K mutant has enhanced membrane-binding affinity [61]. On the other hand, the overexpression of A30P, which has reduced membrane-binding affinity, does not show defects during exocytosis [41]. Taken together, these findings suggest that the membrane-binding capacity of the N-terminal region of αSyn is important for synaptic function. The C-terminal region of αSyn has been shown to bind directly to synaptobrevin-2 (VAMP2), a vesicle-associated soluble NSF-attachment protein receptor (v-SNARE), and αSyn is known to bind to the synaptic plasma membrane via VAMP2, promoting vesicle fusion and clustering [63,64]. In addition, αSyn has been reported to promote the expansion of fusion pores during exocytosis [65]. In addition, other studies have suggested that the binding of αSyn oligomers to the N-terminal region of VAMP2 inhibits toxic αSyn aggregation on synaptic vesicles [66]. These data indicate that αSyn may take different forms and function differently under different conditions.

The binding of αSyn to lipid membranes has been investigated using artificial liposomes in vitro [67]. Negatively charged phospholipids, such as phosphatidylserine (PS) and phosphatidic acid (PA), are known to affect this binding, as are lipids such as sphingolipids and fatty acids [61,68,69,70,71,72,73]. Phospholipids are components of biological membranes but are also important in cellular functions [23]. Previous studies have reported that the neutral phospholipids phosphatidylcholine and phosphatidylethanolamine do not interact with αSyn alone but with vesicles containing acidic phospholipids, such as PA and PS [67,69,72,74]. Interestingly, the lipids found to interact with αSyn in these biochemical experiments resemble synaptic vesicles in vivo and are abundant in PA and PS [75,76]. In addition to the head group, the polyunsaturation of acyl chains is also important for the binding of αSyn to the membrane, which influences the fluidity of the plasma membrane, and short saturated acyl chains have been reported to affect αSyn aggregation [68,72,77]. Moreover, previous studies have reported the interaction between lipid rafts and αSyn. The localization of αSyn at the synapse depends on its interaction with lipid rafts. Experiments using mouse brains have showed that the changes in the composition of lipid rafts and the binding affinity of αSyn may impair the localization of αSyn and the normal function of αSyn at the synapse [41]. The process by which wt αSyn, which is initially nontoxic, aggregates and gains toxicity is unclear at present, but lipids may be involved in this process; one hypothesis is that lipids induce a conformational change in αSyn that leads to the aggregation of αSyn [43]. The other hypothesis is that the lipid membrane promotes an increase in the local concentration of αSyn, which triggers its aggregation. Although αSyn is natively unfolded, upon binding to lipid membranes, the N-terminal region of αSyn undergoes a conformational change from a random coil to an α-helix structure [78,79,80]. However, it is yet unclear as to how membrane-bound αSyn gains a β-sheet structure and initiates aggregation.

In the pathogenesis of PD and DLB, αSyn is thought to change to a β-sheet-rich structure during the process of aggregation, which eventually results in LB formation [81]. Several studies have suggested the prion-like propagation of αSyn aggregates, in which αSyn takes on a pathological structure [82,83,84,85,86,87,88,89,90]. However, it remains unknown as to how αSyn, which is normally considered to be nontoxic, gains such a pathological structure. We hypothesize that the interaction of αSyn with lipid membranes may be one of the key steps in this process. One reason is that all the point mutations linked to familial PD are located in the N-terminal membrane-binding domain of αSyn. However, in vitro studies have shown an inconsistency in the binding affinity of these mutants to lipid membranes. For example, the A30P, A53E, and G51D mutants were reported to have a decreased affinity for lipid membranes compared with wild-type αSyn, the A53T and H50Q mutants had an unchanged affinity, and the E46K mutant had an increased affinity [41,61,69,74,91,92,93,94,95,96,97]. In addition, the A53T and E46K mutants were found to form fibrils with different structures in the absence of membranes [95], suggesting that lipid membranes affect the aggregation process of αSyn. The mechanisms of the conflicting effects of different αSyn mutations on lipid binding and aggregation of αSyn have not been fully elucidated (Table 1). Thus, further studies are needed to understand the relationship between the effects of different αSyn mutations on its lipid binding and aggregation. The second reason is that αSyn has been reported to change its structure from a random coil to an α-helix-containing conformation upon binding to lipid membranes, although this conformational change in αSyn was less efficient in the A30P mutant, which has a reduced affinity for lipid membranes [69]. Third, we demonstrated using Fourier-transform infrared microspectroscopy that lipids accumulate in the central region of LBs in the brains of PD patients [34]. Recently, using high-resolution microscopy, LBs in the brains of PD patients were shown to contain membrane fragments, vesicles, and organelles such as mitochondria and lysosomes [98]. Taken together, it is likely that the interaction between αSyn and lipid membranes is involved in the process of αSyn aggregation and LB formation.

It is also known that the binding of αSyn to membranes can adversely affect membrane integrity and cause deleterious effects. αSyn aggregates, particularly annular oligomers, induce membrane permeabilization like other amyloidogenic proteins such as amyloid-β (Aβ) and tau [99]. The protofibrillar form of αSyn, but not the monomeric and fibrillar forms, was shown to permeabilize synthetic vesicles in vitro [100]. The A30P and A53T mutants of αSyn, but not the G51D mutant, showed higher permeabilizing activities than wt αSyn, suggesting that the effects of mutation on membrane affinity do not always correlate with their permeabilizing activities [101,102].

The N-terminal region of αSyn is amphiphilic and is involved in the interaction with lipid membranes [36,103]. The KTKEGV repeat motif in the N-terminal region of αSyn is well conserved among species [104]. Interestingly, similar amphiphilic [DE]-[DE]-X-R-X-R-L-G repeat motifs are also found in apolipoproteins involved in lipid metabolism [36,105]. In fact, αSyn has similar characteristics to apolipoproteins in that it has amphipathic helices, is inserted into membranes, and affects membrane curvature [106], implying that αSyn may be a potential member of the apolipoprotein family [107]. Several apolipoproteins have been implicated in neurodegenerative diseases [108,109,110]. For example, previous studies have shown that increased levels of apolipoprotein E (ApoE) and its receptor LRP-1 are risk factors for PD. In particular, ApoEε4, which binds to lipids with a higher affinity than other ApoE isoforms, is the most pathogenic and promotes the aggregation of αSyn [111,112]. Paslawski et al. showed that αSyn and apolipoproteins colocalize on lipoprotein vesicles in cerebrospinal fluid, and that ApoE levels were increased in the cerebrospinal fluid of patients with early PD [113]. Considering the similarity between αSyn and apolipoproteins, αSyn may also be involved in the pathogenesis of PD via lipid interactions. As described above, missense mutations in αSyn that cause fPD are all located within and between the KTKEGV repeat motifs in the N-terminal region. These mutations have been shown to alter the interaction between αSyn and lipid membranes, resulting in the aggregation of αSyn.

Recently, protein liquid–liquid phase separation (LLPS) has received much attention as the potential trigger of pathological protein aggregation observed in neurodegenerative diseases. LLPS is a phenomenon in which a protein solution is separated into different liquid phases without mixing, resulting in the formation of liquid droplets. In cells, proteins as well as RNAs are concentrated by LLPS, which contributes to the formation of membraneless organelles, such as nucleoli, stress granules, and P-bodies [114,115,116,117,118,119,120]. Recently, αSyn, as well as other neurodegenerative disease-causing proteins, including FUS, TDP-43, and tau, have been shown to reversibly form liquid droplets by LLPS and to eventually form irreversible amyloid-like fibrils [121,122,123,124,125], suggesting that the liquid droplets formed by LLPS may serve as a precursor of pathological protein aggregation. αSyn is known to have two low-complexity domains (LCDs), which are prone to undergo LLPS, in the N-terminal and NAC regions. Previous studies have shown that both regions are important for the LLPS of αSyn. Moreover, Ray et al. have shown that the formation of liquid droplets of αSyn leads to the formation of its amyloid-like fibrils under various conditions [126]. The formation of liquid droplets and amyloid-like fibrils of αSyn is accelerated by pH changes, high concentrations, phosphate modifications, fPD mutations, metal ions (Fe^3+^, Cu^2+^, and Ca^2+^), and liposomes [126,127]. These results suggest that lipids could affect not only the LLPS of αSyn but also its aggregation. Interestingly, it has been suggested that the central region of LBs contains a large amount of lipids such as sphingomyelin [34,128]. Considering these findings, lipids may play an important role in triggering the aggregation of αSyn in the pathogenesis of PD (Figure 2).

## 3. αSyn and Lipids in Brain Tissue

The classical LBs found in the substantia nigra of PD patients are typically spherical intracellular inclusions that are eosinophilic. They have a dense core in the center, surrounded by a halo of radial fibrils that are approximately 10 nm in width [4,129]. On the other hand, cortical LBs found in the cortex of DLB and advanced PD patients are somewhat distinct and have no halo, unlike classical LBs [4,130]. The main component of LBs is αSyn. Other than αSyn, they are composed of neurofilaments [131], microtubule-associated protein 1B [132], and the galectin-3 [133] proteins, as well as various lipids [34].

In addition to the proteinaceous constituents, there have been reports on the presence of lipids in the LBs of PD patients. Histochemical lipid staining demonstrated the presence of phospholipids, particularly sphingomyelin, in the core of LBs [134]. Another study also reported positive staining with the lipid-soluble fluorescent dye rhodamine B in the core of LBs of PD patients [135]. Using Fourier-transform infrared microscopy, we found an accumulation of lipids that was surrounded by a halo rich in β-sheet fibrils in the core region of LBs in the brains of PD patients [34]. These observations described above are consistent with those of a recent study that used advanced correlative light and electron microscopy to demonstrate that LBs contain αSyn fibrils as well as membranous materials, such as various vesicles, mitochondria, and lysosomes [96].

## 4. Mutations in Lipid Metabolism-Associated Genes and PD Pathogenesis

The involvement of lipids in αSyn aggregation in the pathogenesis of PD is not only supported by the abovementioned lines of biochemical evidence but also by lines of genetic evidence. The GBA1 gene, which encodes the lysosomal enzyme glucocerebrosidase (GCase) and whose homozygous mutations cause Gaucher disease, is recognized as the strongest genetic risk factor for sPD [136,137]. GCase hydrolyzes glucosylceramide (GlcCer), which accumulates in Gaucher disease. To elucidate the mechanism by which mutations in the GBA1 gene increase the risk of developing PD, we conducted a combination of in vitro and in vivo analyses using a *Drosophila* model of PD expressing αSyn [138]. We found that knockdown of GBA1 accelerates the degeneration of dopamine neurons, resulting in motor dysfunction in these PD flies [28]. Furthermore, we found that the GBA1 substrate GlcCer directly acts on αSyn and accelerates the accumulation of proteinase K-resistant αSyn in vitro, suggesting that GlcCer promotes the toxic structural conversion of αSyn [28]. In addition, GlcCer has been shown to colocalize with αSyn in iPS-derived neurons [139]. Moreover, pathological studies have suggested a broad association between αSyn pathology and other lysosomal storage diseases in which other glycosphingolipids accumulate in brains [140,141], implying that such glycosphingolipids may also trigger αSyn aggregation.

Interestingly, decreased activity of the lysosomal enzyme GCase has been reported in the brains of sporadic PD/DLB patients [142,143]. In addition, lysosomal enzymes show high activity in the substantia nigra, and decreases in these activities have been noted in PD/DLB patients and in older individuals [144,145]. Taken together, these findings suggest that lysosomal enzyme activity may be involved in the etiology of PD/DLB.

Mutations in the SYNJ1 gene which encodes the phosphoinositide phosphatase SYNJ1 have been found to be associated with early-onset fPD (PARK20) [146]. Recently, we investigated the role of SYNJ1 in the pathogenesis of PD in relation to αSyn aggregation and found that SYNJ1 deficiency causes the accumulation of phosphatidylinositol-3,4,5-trisphosphate (PIP3), which directly interacts with αSyn and accelerates the formation of αSyn fibrils. Interestingly, these αSyn amyloid-like fibrils are similar in shape to the αSyn amyloid-like fibrils found in PD brains. Furthermore, using a *C. elegans* model of PD, SYNJ1 haploinsufficiency was found to accelerate αSyn accumulation and to induce locomotor defects. Moreover, we confirmed that PIP3 and αSyn colocalize in sPD brains [29]. These results suggest that the interaction of PIP3 with αSyn may contribute to the aggregation of αSyn in the pathogenesis of PD.

Moreover, other genetic variants, including *PLA2G6* (*PARK14*), *VPS13C*, *C19orf12*, and *GALC*, also suggest an association between lipid abnormalities and PD [30,31,32,33]. The *PLA2G6* gene is responsible for an autosomal recessive form of PD (*PARK14*), as well as infantile neuroaxonal dystrophy and neurodegeneration with iron accumulation in the brain, all of which show LB pathology [30]. PLA2G6 belongs to the phospholipase PLA2 family of proteins, which hydrolyze phospholipids and generate free fatty acids and lysophospholipids. PLA2G6 knockout mice were reported to show increased levels and the accumulation of αSyn in their neurons [147]. The loss of iPLA2-VIA, a homolog of PLA2G6, in *Drosophila* resulted in increased ceramide levels and resulted in a shortened lifespan and impaired synaptic transmission due to neurodegeneration [148]. Knockdown of Vps13, which is related to a lipid transport protein, in *Drosophila* was shown to increase αSyn oligomerization [149]. Mutations in the *C19orf12* gene, which is involved in fatty acid supply, has also been reported to be associated with PD [32]. Krabbe’s disease, an inherited lysosomal storage disorder (LSD) caused by mutations in the lysosomal enzyme GALC, is also characterized by neuronal αSyn aggregates [150]. An experimental study demonstrated that a mouse model of Krabbe’s disease exhibits αSyn aggregation in the brain, altered lipid membrane dynamics, and impaired synaptic function and macroautophagy [33].

Gaucher disease is one of the LSDs in which the activity of specific lysosomal enzymes is defective, resulting in the accumulation of their substrates, such as specific lipids, glycoproteins, and mucopolysaccharides, within lysosomes. Interestingly, αSyn accumulation is found not only in the brains of GD patients but also in the brains of several other LSD patients [151,152,153]. For example, in β-galactosialidosis, GM1 ganglioside (GM1) accumulates in the brain, and GM1 was found to specifically bind to αSyn, promoting its aggregation [153]. We also showed that the knockdown of β-galactosidase promotes the formation of proteinase K-resistant αSyn aggregates and worsens locomotor dysfunction in *Drosophila* [28]. These results suggest that accumulated glycolipids interact with αSyn to promote its toxic conversion and subsequent neurodegeneration. It is noteworthy that GM1 also binds directly to the Aβ peptide and induces its toxic conversion [140]. In light of these findings, GM1 may be involved in the toxic conversion of disease-associated proteins to pathological structures that are common among these disease-associated proteins. Ganglioside levels were reported to be increased in the neurons of LSD patients, and the LB-positive LSD patients showed an accumulation of GlcCer, the gangliosides GM1, GM2, GM3, and SM, and cholesterol [141]. Biochemical studies showed the direct binding of αSyn to these lipids, and the binding affinities were as follows: GM3 > Gb3 > GalCer > GM1 > sulfatide > LacCer > GM4 > GM2 > asialo-GM1 > GD3, suggesting that glycolipids with 1, 3, 5 sugar units are preferred [154]. Grey et al. reported that GM1 and GM3 promote αSyn aggregation, and these glycolipids are rich within exosomes, implying that the nucleation of αSyn aggregation is triggered within exosomes [155]. Moreover, saturated fatty acids (SFA), monounsaturated fatty acids (MUFA), and polyunsaturated fatty acids (PUFA) are suggested to be an important factor in αSyn aggregation [156,157,158,159,160,161,162,163,164,165,166]. In addition, the modifications to lipids could also affect the αSyn aggregation process. For example, it has been suggested that lipid peroxidation is involved in the oligomerization of αSyn [167]. These multiple factors may be differentially involved in the impairment of the normal function αSyn. Taken together, altered lipid metabolism may cause the accumulation of lipids in the brain, and these accumulated lipids may promote αSyn aggregation, leading to neurodegeneration.

## 5. Conclusions

In this review, we focused on lipids as one of the key factors in triggering αSyn aggregation. Biochemical studies demonstrated the interaction of αSyn with lipids and the effects of these lipids on αSyn aggregation, and genetic studies also support this concept. Pathological studies also indicated the involvement of lipids in LBs in the brains of PD patients. Thus, we proposed here that the interaction of αSyn with lipids would be one of the key steps that triggers and/or induces αSyn aggregation, although the detailed mechanisms by which lipids promote αSyn aggregation remain to be elucidated. Therefore, we propose that modulating the interaction of lipids with αSyn using small molecules or by modulating the contents of lipids in food, for example, would be a new therapeutic option towards developing potential therapies for PD.

## Figures and Tables

**Figure 1 ijms-25-08935-f001:**
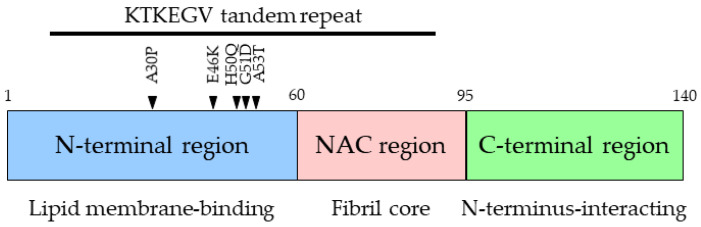
Schematic diagram of three regions of the αSyn protein. The N-terminal region binds to lipid membranes (blue). The non-amyloid-β component (NAC) region is the core of amyloid-like fibrils (red). The C-terminal region interacts with the N-terminal and NAC regions (green). The N-terminal region has tandem repeat motifs consisting of the KTKEGV sequence. Arrowheads indicate αSyn mutations responsible for fPD.

**Figure 2 ijms-25-08935-f002:**
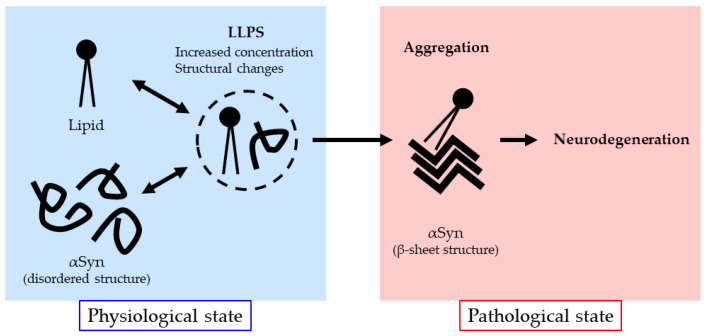
Schematic model of the aggregation process of αSyn. Lipids interact with αSyn and possibly modulate its LLPS-induced liquid droplet formation, which may eventually trigger its aggregation.

**Table 1 ijms-25-08935-t001:** Relationship between αSyn mutation, lipid-binding affinity, and aggregation.

αSyn Mutations	Lipid-Binding Affinity	Aggregation
A30P	↓	↓
E46K	↑	↑
H50Q	≈	↑
G51D	↓	↓
A53T	≈	↑
A53E	↓	

## Data Availability

No new data were created or analyzed in this study. Data sharing is not applicable to this article.

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
