# Peer review of "Role of Lipids in the Pathogenesis of Parkinson’s Disease"

_ijms, 2024, doi:10.3390/ijms25168935_

Round 1
Reviewer 1 Report
Comments and Suggestions for Authors
This review paper aims to explore the role of lipids in the pathogenesis of Parkinson's disease (PD), focusing on their interaction with α-synuclein (αSyn) and its aggregation process.
The main contributions include summarizing current knowledge on αSyn-lipid interactions, discussing genetic evidence linking lipid metabolism genes to PD risk, and proposing lipid modulation as a potential therapeutic approach.
The paper's strengths lie in its overview of αSyn structure, function, and aggregation in relation to lipids, as well as its integration of biochemical, genetic, and pathological evidence to support the hypothesis that lipids play a crucial role in PD pathogenesis.
The authors have effectively identified gaps in knowledge, particularly concerning the mechanism by which αSyn aggregates in vivo. Including recent findings on Lewy body (LB) composition and LB spherical structure.
One area that could be strengthened is the discussion of potential therapeutic approaches based on modulating αSyn-lipid interactions.
Additionally, the review could benefit from
1) A more critical analysis of conflicting findings regarding the effects of different αSyn mutations on lipid binding and aggregation.
2) A figure showing the alignment of KTKEGV motifs between αSyn and selected apolipoproteins.
3) Address if GBA1 deficiency (GCase) leads to Gaucher disease due to accumulation of glycolipid glucocerebroside (also known as glucosylceramide). Line 227: How does GCase hydrolysis of glucosylceramide (GlcCer) leads to accumulation in Gaucher disease?
4) Missing citation for line 94: E46K mutant has enhanced membrane-binding affinity..
5) Missing citation for line 104: N-terminal region of VAMP2 inhibits SNARE-mediated vesicle docking…
Reviewer 2 Report
Comments and Suggestions for Authors
The pathogenesis of Parkinson's disease (PD) is largely dependent on the aggregation of α-synuclein (αSyn) and its accumulation as Lewy bodies, which the authors of this review manuscript examined. It talks about the unknown processes that lead to αSyn aggregation and shows how αSyn interacts with lipids to affect aggregation. Mutations in genes related to lipid metabolism, such as synaptojanin 1 (SYNJ1) and glucocerebrosidase (GBA1), have been associated with PD in both familial and sporadic cases, according to genetic studies. The review highlights the potential therapeutic strategy of treating Parkinson's disease (PD) by modulating lipid interactions with αSyn. Overall, this manuscript combined pathological, genetic, and biochemical evidence to suggest that lipids play a major role in PD pathogenesis through αSyn aggregation. With its innovative approach, this manuscript is highly novel in the field. For the authors' reference, I only have a few remarks from minors.
1. The authors of this review manuscript appear to have dedicated more space to examining how lipid behavior may impact αSyn aggregation. Some of the factors that are thought to act as triggers for αSyn aggregation include the way αSyn interacts with lipid membranes and LLPS associated with LD. On the other hand, since there are a variety of lipids (saturated fatty acids, MUFA, or PUFA), it is unclear which specific mechanisms are involved in αSyn aggregation and whether specific lipid types contribute to it. More information on this topic, perhaps in the form of a table, would be helpful to the field's understanding if the writers could provide it.
2. It appears that the lipid raft plays a significant functional role in the interaction between the lipid membrane and αSyn. Thus, the authors might think about discussing its connection to αSyn in the discussion.
3. This manuscript appears to be centered around the direct interaction between lipids and αSyn. Nevertheless, there exist certain additional factors that impact the impact of lipids on αSyn aggregation that have not been addressed thus far, such as lipid peroxidation resulting from anomalous lipid metabolism. The writers may also think about bringing up these incidental factors during the conversation.
4. The value of this manuscript will be further enhanced if some perspective concepts—particularly potential recommendations for clinical treatment based on the claims of this manuscript—can be added to the final section.
Comments on the Quality of English Languageonly minor editing of English language is required.
